# Confocal Laser Endomicroscopy for Detection of Early Upper Gastrointestinal Cancer

**DOI:** 10.3390/cancers15030776

**Published:** 2023-01-26

**Authors:** Wei Han, Rui Kong, Nan Wang, Wen Bao, Xinli Mao, Jie Lu

**Affiliations:** 1Department of Gastroenterology, Shanghai Tenth People’s Hospital, Tongji University School of Medicine, Shanghai 200072, China; 2Department of Gastroenterology, Taizhou Hospital of Zhejiang Province Affiliated to Wenzhou Medical University, Zhejiang 317099, China; 3Department of Gastroenterology, Gongli Hospital of Shanghai Pudong New Area, Shanghai 200135, China

**Keywords:** CLE, early oesophageal cancer, early gastric cancer, Barrett’s esophagus

## Abstract

**Simple Summary:**

Early diagnosis of digestive tract neoplasms is beneficial for maximising the possibility of cure. The diagnostic efficacy of conventional endoscopy is currently limited. Confocal laser endomicroscopy (CLE) can magnify tissue to the cellular level, which is beneficial for endoscopic biopsy. Currently, its clinical application has attracted wide interest among endoscopists. Many studies have explored its application potential in the early diagnosis of digestive tract tumours. This paper reviewed a large number of clinical studies on CLE in early esophageal and gastric cancers, providing more evidence-based support for endoscopists in making clinical decisions.

**Abstract:**

Esophageal and gastric cancers are common diseases with high morbidity and mortality; thus, early detection and treatment are beneficial to improve prognosis. Confocal laser endomicroscopy (CLE) is a novel imaging technique that permits the histological analysis of tissues during endoscopy. CLE has been shown to uniquely affect the diagnosis of early upper gastrointestinal cancers. Relevant literature was searched using PubMed and Google Scholar databases. Despite inherent flaws, CLE can reduce tissue damage and improve diagnostic accuracy to a certain extent. CLE in combination with other imaging methods can help enhance the detection rate and avoid unnecessary biopsies in the management of esophageal or gastric cancer and precancerous lesions. CLE is of great significance in the diagnosis and surveillance of early cancers of the upper gastrointestinal tract. Further technical innovations and the standardisation of CLE will make it more responsive to the needs of routine clinical applications.

## 1. Introduction

Owing to high morbidity and mortality levels, upper gastrointestinal cancers have become severe malignant diseases that threaten people’s health worldwide. In the latest global cancer statistics report in 2020, the incidence of esophageal cancer was 3.1% (seventh place), the mortality rate was 5.5% (sixth place), the incidence of gastric cancer was 5.6%, ranking fifth, while the mortality rate was 7.7%, the fourth highest among all cancers [1]. The early diagnosis of upper gastrointestinal cancers and precancerous lesions is of great value for their treatment and prognosis. Traditional white light endoscopy (WLE) has many limitations in the observation of microscopic lesions. The evolving optical-based endoscopy technology offers the potential to detect the earliest mucosal changes at microstructural, biochemical, and molecular levels [2]. Confocal laser endomicroscopy (CLE), which has increased in recent years, can be used for the histological analysis of tissue during endoscopy, also known as optical biopsy. CLE is based on the principle of illuminating the mucosal surface with a blue laser and detecting the fluorescence reflected from the tissue. The laser is focused at a specific depth and only the light reflected from the plane is refocused through the pinhole confocal hole. Therefore, the image resolution is enhanced [3,4]. CLE can magnify the mucosal structure by a factor of 1000, making it possible to view the structure of the mucosa in real time at the cellular or subcellular level. Two sets of CLE systems can be used for gastrointestinal endoscopy. One of the first systems developed was eCLE, which integrated a miniature confocal scanner into the tip of a special flexible endoscope. However, this system is rarely used clinically. Another system, the pCLE, has a flexible confocal microprobe that can pass through the working channels of most traditional endoscopes. Compared with eCLE, it can acquire images faster, but its resolution and preset fixed plane depth are limited [3,5]. Needle-based CLE (nCLE) has been the latest development of CLE in the past few years. Essentially, a CLE miniprobe passes through the 19th endoscopic ultrasound (EUS)-FNA needle, which achieves the effect of optical needle biopsy. nCLE provides 500–1000× real-time microscopic imaging of living tissues, which facilitates the study of pathological entities in their natural environments during functional imaging [6]. The main application of nCLE is the evaluation of pancreatic cystic neoplasms and lymph nodes. The application of nCLE in the detection of gastrointestinal subepithelial lesions (SELs) has also been reported. The advantages of CLE are that it increases the resolution and contrast of optical imaging and simultaneously realises in vivo imaging to avoid artefacts caused by tissue processing. The esophageal epithelium, mucosal capillary network, gastric pit, glandular structure, and goblet cells can be easily observed [7]. CLE plays a unique role in the detection of early upper gastrointestinal cancers. This article reviews the applications of CLE in early esophageal cancer, early gastric cancer, and their precancerous lesions.

## 2. Esophageal

### 2.1. Barrett’s Esophagus (BE) and Esophageal Adenocarcinoma

Barrett’s esophagus (BE) with intestinal metaplasia (IM) or dysplasia is the main precancerous lesion in esophageal adenocarcinoma. Endoscopic treatment of esophageal adenocarcinoma in the early stages of its development (dysplasia and intramucosal carcinoma) is of paramount importance for patient prognosis, so endoscopic surveillance of BE is essential [8]. Current BE surveillance still uses high-definition WLE (HDWLE) with random four-quadrant biopsies every 2 cm and targeted sampling of visible lesions (Seattle protocol) [9,10]. However, some studies have reported defects in this method. For example, long-term surveillance requires a large number of biopsies that may cause tissue damage, and the accuracy of biopsy diagnosis is sometimes unsatisfactory, especially in patients with high-grade dysplasia (HGD) [11,12]. Therefore, some guidelines no longer recommend this method [13,14]. CLE may provide a new solution to these problems. Mainz confocal Barrett’s classification was first developed by Kiesslich et al. to predict BE or BE-associated neoplastic changes under eCLE [15]. Canto et al. used the Mainz classification to compare the diagnostic abilities of HDWLE with random biopsy (RB) and HDWLE plus eCLE with targeted biopsy (TB) in a multicentre international randomised controlled trial [16]. The study concluded that the sensitivity, specificity, positive predictive value (PPV), and negative predictive value (NPV) of HDWLE + eCLE with the targeted biopsy group are 95%, 92%, 77%, and 98%, respectively, for a per-patient analysis. eCLE not only effectively improved the sensitivity (40% for HDWLE + RB vs. 95% for HDWLE + eCLE, *p* < 0.0001) of dysplasia diagnosis but also reduced the number of biopsies (median number of biopsies per patient: two for HDWLE + eCLE vs. four for HDWLE alone, *p* < 0.0001). However, this system is no longer commercially available, and there have been few randomised controlled trials on eCLE in recent years; therefore, whether eCLE can truly replace standardised biopsy protocols requires more clinical evidence. As for pCLE, the Miami criteria is the most widely used classification verified by a large number of randomised controlled trials (Figure 1a,b) [17]. Several studies have reported that pCLE plays an important role in BE surveillance. In an international, prospective, multicentre, randomised, and controlled trial that involved 101 consecutive patients with BE [18], when adding pCLE, the sensitivity (68.3% vs. 34.2%) and NPV (94.6% vs. 89.8%) for the diagnosis of HGD and early carcinoma increased by 34.1% and 4.4%, respectively, with a 4.9% reduction in specificity (87.8% vs. 92.7%) compared with HDWLE alone in the per-location analysis. As for per-patient analysis, the sensitivity, specificity, and NPV of HDWLE and pCLE were 93.50%, 67.10%, and 95.90%, respectively. Another single-centre study by Bertani et al. [19], which was performed outside tertiary referral centres, achieved high sensitivity and NPV for detecting dysplasia of 100%, with a specificity of 83%. Recently, a prospective study compared the accuracy of pCLE and traditional biopsy methods in the diagnosis of BE-related cancer and dysplasia and evaluated the feasibility of the clinical application of pCLE [20]. This study assessed real-time pCLE based on HDWLE and narrow-band imaging (NBI) or blinded pCLE, which was blinded to the endoscopic images and real-time pCLE interpretation. Both real-time pCLE and blinded pCLE interpretation had a low sensitivity for HGD or cancer of 67%; the specificities of the two groups were 98% and 94%, respectively. The NPV was 98% for real-time pCLE and 33% for blinded pCLE. For low-grade dysplasia (LGD), real-time pCLE interpretation also had a low sensitivity of 60% and a specificity of 87%. As for IM, a recent prospective, multicentre study showed that early pCLE endoscopists had higher detection rates of IM than the Seattle protocol overall (99/172 vs. 46/172, *p* < 0.0001) in BE, especially in patients with a visible columnar-lined esophagus (75 vs. 31, *p* < 0.0001) [21]. Endoscopic surveillance after endoscopic treatment had a pivotal role in the persistence and recurrence of dysplasia. A single-centre, prospective, pathologist-blinded study explored the value of pCLE in detecting persistent/recurrent IM/neoplasia after endoscopic treatment [22]. The results showed that pCLE and ordinary biopsy had almost the same sensitivity, specificity, PPV, and NPV in diagnosing recurrent/persistent IM. This is the first study on pCLE to diagnose postoperative recurrence of IM after endoscopy treatment, and its potential in this area has yet to be assessed in a large sample size. For the past few years, the continuous update based on the CLE diagnostic criteria has been conducive to further improving the diagnostic accuracy for BE-related cancer stages. Krajciova et al. proposed an automatic BE pathology stage classification system based on support vector machine (SVM) [23], and the system achieved a satisfactory sensitivity of 97% and specificity of 96% for diagnosing IM in a database of 557 images from 96 patients. Moreover, its sensitivity and specificity for detecting dysplasia reached 94% and 97%, respectively. Another study focused on the endoscopic features of pCLE in BE-related LGD [24]. They selected six microscopic features as positive indicators and met three of the six items as diagnostic criteria. In phase II, the sensitivity and specificity for diagnosing BE-related LGD were 81.9% and 74.6% on a set of 57 pCLE videos, respectively. There are some differences in the results due to factors such as the number of participants, study design, and proficiency of the endoscopist. Some meta-analyses have conducted a summary analysis of these studies [25,26,27,28,29]. These analyses yielded similar results (Table 1). The Preservation and Incorporation of Valuable Endoscopic Innovation (PIVI) standard established by the American Society of Gastrointestinal Endoscopy emphasises that imaging technology with targeted biopsies should achieve a per-patient sensitivity of 90% or higher, an NPV of 98% or higher, and a high specificity of 80% to decrease the number of biopsies for detecting HGD or early esophageal adenocarcinoma compared with the Seattle protocol [25]. According to the white paper AGA, which provides suggestions for the application of enhanced imaging techniques, experienced endoscopists can use new imaging techniques, such as CLE, if they meet the thresholds recommended in the ASGE PIVI. Otherwise, a four-quadrant biopsy must be added to BE surveillance [30]. In most per-patient studies, the specificity of CLE can reach more than 80%, but its sensitivity is still less than 90%, which indicates that it is unable to be used alone for BE monitoring in most medical centres. In addition, evidence-based quality indicators must be formulated to help endoscopists master and proficiently apply CLE imaging technology. Owing to some defects in CLE technology, it is difficult to apply it to the routine management of BE and BE-related esophageal carcinoma independently. However, under the premise of other technologies as the red flag, CLE can be used as a powerful supplement to improve diagnostic accuracy. Pietro et al. tested the diagnostic value of autofluorescence imaging (AFI) combined with pCLE for dysplasia and intramucosal carcinoma [31]. Fifty-five selected patients were sequentially subjected to magnified WLI and AFI examinations, and then AFI-targeted areas (AFI (+) and one AFI (−) control) underwent NBI, pCLE, and biomarkers on biopsies. Random biopsies were performed according to the Seattle protocol for histopathological analysis. The results showed that AFI-targeted pCLE had satisfactory sensitivity for detecting any grade of dysplasia, and the sensitivity was equal to that of the Seattle protocol (96.4% vs. 92.8%). The average number of biopsies required for AFI-targeted pCLE was only one-third that of the standard clinical protocol (3.5 per-patient vs. 12.9 per-patient). In addition, the study concluded that there was a correlation between the number of positive biomarkers and the degree of dysplasia. Combined AFI-targeted pCLE and molecular markers can increase the diagnostic specificity of any grade of dysplasia from 74.1% to 88.9%. It can also guide clinical strategies; patients who are diagnosed with dysplasia under optical imaging and have at least one positive molecular marker should receive active clinical treatment. This study provides new insights. Based on other novel imaging technologies, such as AFI as the red flag, pCLE can give full play to its advantages and effectively improve the sensitivity for the diagnosis of dysplasia and esophageal adenocarcinoma. This will facilitate early intervention in patients with high-risk BE. The combination of molecular markers with optical biopsy further reduces false positive rates, thereby avoiding unnecessary biopsies and treatments. 

### 2.2. Early Squamous Neoplasms

Early esophageal squamous cell carcinoma (ESCN) is classified into low-grade intraepithelial neoplasia (LGIN), high-grade intraepithelial neoplasia (HGIN), and superficial ESCN. Lugol’s chromoendoscopy (LCE) is the most common method for detecting ESCN. Although it has high diagnostic sensitivity, some drawbacks affect its performance. For example, iodine spray causes discomfort to patients, and lower diagnostic specificity results in the need for more biopsies [32,33,34,35]. CLE has a unique effect on the diagnosis of ESCN, with the advantage of depicting cell and capillary structures in real time (Figure 1c,d). Liu et al. established a cellular and intrapapillary capillary loop (IPCL) pattern criteria for diagnosing ESCN by comparing the different characteristics of squamous cells and IPCLs in normal epithelium and malignant esophageal lesions [36], which achieved a sensitivity of 94.1% and a specificity of 100%. Several studies have reported that CLE could be used as an adjunctive method for predicting ESCN based on cellular and vascular criteria. In a single-centre study [37], chromoendoscopy-guided eCLE was compared with LCE for the detection of ESCN. eCLE revealed 45/56 lesions with neoplasms and 11/56 lesions with non-neoplasms. After the histological examination of en bloc ER specimens, the diagnosis of 44/45 for neoplasms and 9/11 for non-neoplasms was proven correct, which obtained the sensitivity, specificity, and NPV of 95.7%, 90.0%, 81.8%, respectively. Compared with those of LCE alone, the sensitivity (89.1% vs. 95.7%) and NPV (64.3% vs. 81.8%) of chromoendoscopy-guided eCLE were significantly improved, with the same specificity (90%) as LCE. Another single-centre, non-randomised, cross-sectional trial by Pech et al. explored the diagnostic value of pCLE and dual-focus NBI (dNBI) in 43 Lugol’s voiding lesions > 5 mm [38]. In seven cases diagnosed by histology of ESCN, pCLE had higher specificity (92% vs. 62%), PPV (83% vs. 54%), NPV (92% vs. 89%), and accuracy (89% vs. 70%) than dNBI, although it was not statistically significant. However, the diagnostic specificity of Lugol’s spray combined with pCLE was three times as high as that of the iodine spray technique alone (92% vs. 33%). Based on this literature, we are not sure whether pCLE is superior to dNBI in the diagnosis of ESCN. Nevertheless, pCLE can serve as a complementary method to reduce unnecessary biopsies of Lugol’s voiding lesions. In a recent article, Piyapan et al. further investigated the role of dNBI, pCLE, and LCE in the surveillance of ESCN in patients with a history of head and neck cancer [39]. This study showed the additional benefit of LCE combined with pCLE in detecting dNBI-missed lesions, especially LGIN. LCE combined with pCLE revealed two HGIN and three LGIN missed by dNBI, which achieved a sensitivity, specificity, and NPV of 80%, 67%, and 91%, respectively, in lesions not detected by dNBI. This demonstrates the feasibility of using dNBI as a first-mode examination method and LCE combined with pCLE as an additional method in clinical applications, which not only improves diagnostic sensitivity for LGIN but also avoids excessive biopsy. In summary, LCE remains the first-line method for diagnosing ESCN. However, the addition of CLE is beneficial for acquiring targeted biopsies to improve diagnostic specificity. In addition, if CLE reveals a benign pattern in Lugol’s voiding lesions, biopsy for histological analysis is unnecessary. Li et al. used CLE to perform three-dimensional endomicroscopic reconstruction of the esophagus and observed the esophageal epithelium and IPCL to create new 2D confocal endomicroscopic criteria (SMS) based on surface maturation [40]. Comparing the two criteria, SMS had higher sensitivity (81.0% vs. 42.9%) and accuracy (87.5% vs. 67.2%) than irregular IPCL criteria and higher sensitivity (81.0% vs. 42.9%) than heterogeneous dark cell criteria. The differences were statistically significant (*p* < 0.05). Subsequently, they utilised SMS criteria to validate the diagnostic potential of pCLE for esophageal squamous neoplasia [41]. A total of 356 participants first underwent virtual chromoendoscopy (I-Scan). pCLE further observed I-Scan positive lesions, and targeted biopsy of pCLE was obtained for histological analysis. pCLE showed excellent sensitivity (94.6%), specificity (90.7%), and NPV (96.1%) for the diagnosis of the I-Scan suspicious lesions. I-Scan plus pCLE can achieve higher specificity (92.9% vs. 22.9%, *p* < 0.001) and NPV (84.4% vs. 61.5%, *p* < 0.001) than I-Scan alone. The pCLE-targeted biopsy efficiently reduced the number of tissue biopsies required for analysis. Li et al. demonstrated that the SMS classification method avoids the effects of angiogenesis and proliferative state, and the difference between intraepithelial neoplasia and non-neoplastic processes is more pronounced. However, there is only one study on this new classification method, and its diagnostic accuracy remains to be further verified. 

The current technological innovation of CLE is expected to compensate for the inherent defects of CLE, making it more convenient for clinical practice. Spectrally encoded confocal microscopy uses high-speed reflectance imaging technology to allow cytology imaging of a broader range of the esophagus [42]. Some studies have reported the application of this technique for the surveillance of patients with BE. A near-infrared probe-based confocal microendoscope (pCM) is a diagnostic device that permits deep tissue imaging with a laser source of 785 nm, a probe with a diameter of 2.6 mm, and a long working distance. The deep tissue structure, such as the squamous epithelium in the stratum corneum, dense cells in the stratum spinosum, vessels in the third layer, and fine fibres in the lamina propria of the esophagus, can be displayed using near-infrared pCM technology [43]. This is promising for the management of esophageal diseases. Table 2 summarizes studies on the diagnostic value of CLE in the esophagus.

## 3. Stomach

### 3.1. Precancerous Conditions

The pathological progression of gastric cancer mainly involves the following processes: chronic gastritis, chronic atrophic gastritis (AG), gastric intestinal metaplasia (GIM), gastric intraepithelial neoplasia (GIN), and early gastric carcinoma (EGC). AG, GIM, and GIN are common precancerous lesions. Active follow-up of patients with these conditions will contribute to the early detection and intervention of gastric cancer. CLE has some special applications in the surveillance of these diseases (Figure 2). Zhang et al. observed the morphology of gastric pits under different pathological conditions using eCLE and divided the gastric pits into seven types [44]. Normal gastric mucosa appeared as round pits with a round opening and was classified as type A. In AG, the number of gastric pits is reduced and the calibre is significantly dilated (type E). The pits show a villous-like appearance, and black goblet cells can be observed in GIM mucosa (type F). In this classification method, the type E pattern for predicting AG had a sensitivity of 83.6%, a specificity of 99.6%, and an NPV of 97.6%. Mucin in goblet cells is not stained with sodium fluorescein, so GIM is convenient for diagnosis. Wallace et al. proposed the Miami classification system to distinguish between different pathological states using pCLE [17]. For gastric dysplasia, an irregular crypt lumen and dark, irregular, thickened epithelium can be observed under this classification system. Li et al. added an index of vascular structure based on the Miami classification method to facilitate a more comprehensive assessment of gastric mucosa [45]. This new pCLE classification contains three types of pit patterns with seven subtypes (type 1, types 2a–c, types 3a–c) and three types of vessel architecture (type 1, type 2, and type 3). These gastric pit patterns, combined with vascular changes in neoplastic gastric mucosa, can predict LGIN and HGIN. In the next prospective study involving 244 patients, the type 2b pit pattern for detecting AG achieved a sensitivity of 88.51% and specificity of 99.19%. Type 2c pit pattern for detecting GIM obtained a sensitivity of 92.34% and a specificity of 99.34%. The sensitivity, specificity, and NPV for LGIN in offline pCLE were 89.86%, 99.25%, and 99.13%, respectively. The sensitivity, specificity, and NPV for HGIN in offline pCLE were 88.89%, 88.89%, and 99.95%, respectively. There were no significant differences between the results of real-time and off-time diagnosis. This new classification describes the blood vessel changes in the pathological progression of gastric mucosa and details the various pathological types (including LGIN and HGIN), which may be more practical in clinical applications. Several studies have addressed the diagnostic value of CLE for precancerous gastric lesions. Liu et al. compared the abilities for detecting AG between chromoendoscopy (CE), NBI, and eCLE [46]. The results showed that eCLE had higher sensitivity (92.31% vs. 83.85%) and specificity (86.18% vs. 78.86%) than CE, and the difference was significant. Metaplastic AG has distinct features under CLE, such as the presence of goblet cells, brush borders, columnar absorptive cells, and villiform foveolar epithelium [47]. eCLE had satisfactory sensitivity (91.94%), specificity (96.86%), and NPV (97.37%) for the identification of metaplastic AG. This study highlights the advantages of CLE in distinguishing between metaplastic AG and non-metaplastic AG. Another study by Yu et al. further evaluated the diagnostic ability of pCLE for AG [48]. After the phase I pCLE diagnostic criteria were determined, 431 patients underwent pCLE and WLI examinations. The results showed that the sensitivity and specificity of pCLE in the diagnosis of AG were 90.3% and 78.8%, respectively, compared to 95.6% and 41% of WLI. The diagnostic efficacy of CLE compared with that of other novel optical imaging techniques has also been explored. In a prospective study with a small sample by Lim et al. [49], real-time pCLE proved to be more sensitive (90.9% vs. 68.2%, *p* = 0.001) and specific (84.7% vs. 69.5%, *p* < 0.05) than AFI, but not significantly different from NBI. In addition, there was evidence that off-time CLE had higher diagnostic accuracy (95.2% vs. 88.0%, *p* < 0.05) than real-time CLE for diagnosis of GIM, which may be because off-time CLE avoids the effect of scanned image velocity and can view the videos repeatedly. However, the small sample size in this study reduces its credibility, and investigation of the diagnostic efficacy of CLE requires a multicentre and large-sample study. The widely accepted guideline, the updated Sydney System, recommends that the standard biopsy protocol should include five parts: two from the antrum (3 cm from the pylorus, greater/lesser curvature), two from the corpus (one from the lesser curvature, 4 cm proximal to the incisura, one from the middle of the greater curvature), and one from the incisura [50]. Currently, the validity of its diagnosis is being questioned. A prospective, double-blind, randomised trial compared the diagnostic yield of GIM between eCLE with targeted biopsy and WLE with the standard biopsy protocol [51]. A total of 168 patients were divided into two groups. Eighty-five patients underwent targeted biopsy at sites suspected of GIM or neoplasia after accepting eCLE at macroscopic lesions and five standardised locations (group A). In group B, biopsies were obtained from endoscopically macroscopic lesions and five standard biopsy sites following the updated Sydney System under WLE. Although there was no significant difference in diagnostic yield between the two groups in the per-patient analysis, the diagnostic yield (65.70% vs. 15.73%, *p* < 0.001) of group A was 50% higher than that of group B in the per-biopsy analysis. GIM and GIN could be detected with 91.67% sensitivity and 96.77% specificity. In addition, the number of biopsies needed in the eCLE-targeted biopsy group was only one-third that in the WLE-standard biopsy group. CLE has the advantages of a higher diagnostic yield and fewer biopsy requirements, which makes it possible to replace the standard biopsy protocol. Another retrospective study based on outpatient clinics pointed out that the sensitivity of pCLE combined with WLE to diagnose AG or GIM was 86.8%, and the specificity was 81.8%. Although it had a higher sensitivity of 96.3% for the diagnosis of GIN, the diagnostic specificity was still low at 86% [52]. A recent multicentre retrospective study demonstrated the excellent clinical application of pCLE in gastric cancer and precancerous lesions [53]. In their study, the diagnostic sensitivity rate of pCLE for several precancerous lesions, including AG, GIM, LGIN, and HGIN, was greater than 80%, among which the diagnostic sensitivity rate for AG and HGIN was greater than 90%. Moreover, the specificity for HGIN reached 97.17%, and the specificities for the diagnosis of AG and GIM were 91.09% and 92.16%, respectively, whereas the specificity of LGIN was still low, only 87.50%. Many technologically updated measures have further improved the diagnostic efficiency for gastric precancerous lesions. A document proposed that propofol-based sedation further improved the sensitivity of pCLE to diagnose GIM (85.55% vs. 50.00%) and GIN (65.11% vs. 40.00%) [54]. At the same time, the diagnostic specificities of GIM (80.28% vs. 68.75%) and GIN (92.78% vs. 75.39%) in the sedation group were higher than those in the unsedated group. Similarly, Sun et al. proposed to use topical dye cresyl violet to enhance the detection accuracy of pCLE for GIM [55], which showed that the sensitivity, specificity, PPV, and NPV of per-location analysis of pCLE in the diagnosis of GIM were 91.95%, 93.51%, 86.96%, and 96.11%, respectively. This is an alternative dye that can be utilised in pCLE for the detection of gastric precancerous lesions, but its safety and availability still need to be further explored. In recent years, virtual chromoendoscopy technologies have demonstrated the superiority of gastrointestinal tract diseases. Some studies have focused on investigating virtual chromoendoscopy combined with CLE to diagnose and monitor gastric precancerous lesions. In a study involving 55 people who had been diagnosed with GIM [56], the researchers first used 100× magnifying flexible spectral imaging color enhancement (FICE) to observe the gastric mucosa and then performed a pCLE examination at sites suspected of GIM. After the addition of pCLE, there was no significant change in the sensitivity, but the specificity (90.5% vs. 79.2%) significantly increased, and the number of false positive results decreased. This was an earlier study on the feasibility of using virtual chromoendoscopy combined with CLE for the surveillance of gastric precancerous lesions. Recently, a multicentre randomised controlled trial provided a more comprehensive assessment of the diagnostic value of FICE-guided pCLE for the detection of GIM and GIN [57]. A total of 238 patients were divided into the FICE-guided pCLE with targeted biopsy group (group A) and the FICE with standard biopsy group (group B). The results showed the diagnostic yield of group A was significantly higher than that of group B (75.1% vs. 31.5%, *p* < 0.001). In the FICE-guided pCLE with targeted biopsy group, the sensitivity, specificity, PPV, and NPV for predicting LGIN were 87.5%, 98.0%, 87.5%, and 98.0%, respectively, and the sensitivity, specificity, PPV, and NPV for detecting GIM were 95.0%, 94.6%, 90.5%, 97.2%, respectively. In addition, the average number of biopsies required in group A was smaller than that required in group B (3.5 vs. 6.8, *p* < 0.001). The combination of FICE and CLE not only compensates for the shortcomings of FICE, whose specificity diminishes owing to the joint use of multiple criteria, but also solves the problem of small-field microscopic observation of pCLE. Although the current guidelines do not recommend CLE as a routine means of diagnosing and monitoring precancerous lesions of gastric cancer [58], there is no denying that CLE has great potential in the future. CLE may be used based on other imaging techniques, such as NBI and FICE, as the red-flag method, which can help to achieve the target biopsy to reduce the number of biopsies and improve detection accuracy. Further research can continue to explore the feasibility of CLE in conjunction with other virtual chromoendoscopies to maximise the diagnostic value of CLE.

### 3.2. Early Gastric Carcinoma (EGC)

EGC is confined to the gastric mucosa or submucosa and early diagnosis and endoscopic treatment can effectively improve the five-year survival rate. Currently, more advanced imaging technologies are being developed. However, the diagnosis of EGC still relies on histological examination as the gold standard. There are some limitations in that: (1) the histological results between preoperative biopsy and resected tissue after surgery are inconsistent, resulting in inaccurate clinical decision making [59], and (2) repeated biopsy causes mucosal fibrosis, which leads to postoperative complications such as bleeding, perforation, and incomplete resection of the diseased mucosa [60,61]. The use of CLE to perform an optical biopsy may reduce the occurrence of these conditions. The classification of gastric pit patterns by Zhang et al. divided gastric cancer into poorly differentiated tubular adenocarcinoma and differentiated tubular adenocarcinoma [44]. The difference between the two was whether diffusely atypical cells existed during CLE. Li et al. proposed that the vascular manifestations of the neoplastic gastric mucosa were irregular capillaries with expanded calibre or heterogeneous leakage [45]. In the widely used Miami classification [17], EGC appears as a completely disorganised epithelium, fluorescein leakage, and dark irregular epithelium. A study by Bok et al. used the Miami classification to assess the accuracy of conventional biopsy, in vivo pCLE, offline pCLE, and either conventional biopsy or in vivo pCLE in the diagnosis of superficial gastric neoplasia [62]. Compared with traditional biopsy, either conventional biopsy or in vivo pCLE had higher sensitivity (96.9% vs. 75.0%) and NPV (95.7% vs. 73.3%). The results suggested that pCLE can be used as an important complement to traditional biopsy and can replace biopsy or help with targeted biopsy in the case of the endoscopic appearance of highly suspicious superficial gastric neoplasia, thereby avoiding damage to tissues caused by repeated biopsy. However, pCLE does not show a nuclear structure; therefore, it cannot clearly distinguish well-differentiated adenocarcinoma from dysplasia. In addition, it is difficult to detect signet ring cell carcinoma due to the limitation of the observation depth. In response to some shortcomings of the Miami classification, Li et al. conducted a prospective study to conclude a new set of two-tiered CLE classifications for gastric superficial cancer by analysing the eCLE images of 182 patients with stomach disease in phase I [63]. In the two-tiered CLE classification, gastric mucosa falls into non-cancerous lesions or cancer/HGIN lesions according to the different characteristics of the architecture, cells, and microvessels. For cancer/HGIN lesions, glandular size and shape are irregular, pits and glands are disorganised or destroyed, the cells become disordered, stratified, and lose polarity, and the shape and calibre of the microvessels are irregular. In phase II, the diagnostic value was explored using histological results as the gold standard. The sensitivity (88.9% vs. 72.2%), specificity (99.3% vs. 95.1%), PPV (85.3% vs. 41.6%), NPV (99.5% vs. 98.6%), and accuracy (98.8% vs. 94.1%) of eCLE for detecting EGC/HGIN lesions using two-tiered CLE classification were higher than those of WLE. This two-tiered method solved the problem of the unclear classification of LGIN or gastritis and HGIN or gastric cancer according to previous criteria. Using the new classification method, doctors only need to judge whether the patient continues to follow up or undergo endoscopic treatment, which is convenient for doctors to determine the diagnosis and decide the next clinical strategy quickly. Gong et al. compared the diagnostic accuracy of eCLE and ME-NBI for gastric cancerous lesions using the two-tiered CLE classification method [64]. The results showed that the sensitivity (91.67% vs. 90%, *p* = 0.80), specificity (95.45% vs. 93.48%, *p* = 0.68), NPV (93.33% vs. 91.49%, *p* = 0.74), and PPV (94.29% vs. 92.31%, *p* = 0.73) of ME-NBI in the diagnosis of cancer/HGIN lesions were better than eCLE, but there was no statistical significance. In addition, eCLE for distinguishing undifferentiated gastric cancer from differentiated gastric cancer was not statistically significantly better than ME-NBI. The combination of eCLE and NBI can achieve higher sensitivity (94.44%) than NPV (95.24%), but their specificity (90.91%) and PPV (89.47%) were lower than those of NBI or eCLE used alone. eCLE may not be superior to NBI in predicting EGC; however, in some special cases, eCLE can be preferred, such as in mucosal surfaces covered by ulcers, mucosa with bleeding tendency, and suspicious undifferentiated gastric cancer. The conjunctive use of ME-NBI and eCLE does not bring huge benefits, which helps prevent the misdiagnosis of EGC but may increase unnecessary therapy. Recently, a prospective single-centre open-label pilot study examined the ability of CLE to diagnose atypical EGC after Helicobacter pylori eradication [65]. The diagnostic yield (97% vs. 72%, *p* < 0.05) of pCLE for EGC after Hp eradication was higher than that of WLE, and the difference was statistically significant. The diagnostic accuracy (91.7% vs. 69.4%, *p* < 0.05) for the horizontal margin of EGC was significantly higher than that of ME-NBI, although the diagnostic yield (97% vs. 89%, *p =* 0.371) of pCLE was not significantly different from ME-NBI. EGC lesions after Hp eradication lack distinct tumour characteristics and definite boundaries on endoscopy. Even in histological examination, the cancer tissue is always covered by non-neoplastic epithelium, which makes diagnosis difficult. CLE can observe cancerous images under the mucosa covered by non-neoplastic epithelium or mature ambiguous tumours, which makes it more advantageous in the diagnosis and boundary definition of this type of atypical EGC. At present, some studies have reported new applications of CLE in determining EGC boundaries. A prospective, randomised controlled study involving 101 participants compared the ability of pCLE and WLI with chromoendoscopy (CE) to describe the margin of EGC [66]. The investigators divided the 89 lesions into pCLE groups and CE groups. In the two groups, the distal and proximal ends of the tumour were labelled with electrocautery, and an excellent marking rate (the marking dot was less than 1 mm from the histologic tumour margin) and a complete resection rate were determined. The average distance (1.3 vs. 1.8 mm, *p* = 0.525) between the marking dot and the histologic tumour margin in the pCLE group was shorter than that in the CE group. The excellent marking rate (43.9% vs. 27.6%, *p* = 0.023) was also better in the pCLE group, and the difference was statistically significant. In particular, pCLE offered clear superiority for boundary definition of superficial flat lesions, which achieved a significantly shorter average distance (0.5 vs. 3.1 mm, *p* = 0.007) and significantly higher excellent marking rate (72.7% vs. 30.0%, *p* = 0.006) than that of CE. The precise definition of tumour boundaries before ESD has a greater impact on prognosis. As a commonly used boundary definition method in clinical practice, CE has difficulty evaluating superficial flat lesions. The margins of superficial flat EGC determined by pCLE are consistent with the histologic tumour margins after ESD, although there are still many limitations in the application of CLE. In clinical practice, endoscopists can choose different imaging technologies to determine tumour margins according to the characteristics of the diseased mucosa. pCLE is also one of the methods that can be considered. Currently, no guidelines recommend CLE as a routine screening method for EGC. However, a guideline emphasises the importance of reducing the number of biopsies to avoid fibrosis and simultaneously ensuring the diagnostic rate of EGC at the same time [67]. Therefore, CLE-targeted biopsy may be a new approach. In addition, CLE is superior in the diagnosis and boundary definition of some specific types of EGC, such as atypical EGC lesions after Hp eradication, superficial flat EGC lesions, and mucosal surfaces covered by ulcers.

nCLE is a novel diagnostic technique that has the potential to achieve real-time optical biopsy during endoscopic ultrasound. Several studies have investigated the safety and accuracy of EUS-guided nCLE in detecting gastric SELs. SELs include a series of lesions with malignant potential, such as gastrointestinal stromal tumours (GIST), and malignant lesions, such as gastric adenocarcinoma [68]. Zhang et al. obtained the diagnostic criteria for SELs by observing nCLE images and the corresponding histopathology [69]. In a subsequent retrospective review using recorded images, they concluded that offline nCLE had better diagnostic sensitivity and specificity than EUS in detecting GIST and gastric carcinomas, although the results were not statistically different. They proposed the feasibility and safety of EUS-nCLE for SEL detection. In another prospective study including 60 patients by Zhang et al. [70], satisfactory sensitivity (84.2%) and specificity (95.1%) regarding nCLE were observed in the offline diagnosis of GIST; it also had a sensitivity of 100% and specificity of 97.9% for the diagnosis of gastric carcinoma. Its safety was further evaluated, and no EUS-nCLE-related adverse events were observed. Research and exploration of nCLE in gastric SELs remain very limited, and more comprehensive and multicentre research on the diagnostic value of nCLE for SELs is needed to provide additional evidence for its clinical application. Table 3 summarizes studies on the diagnostic value of CLE in the stomach.

## 4. Advantages and Disadvantages

As a novel imaging technique for in vivo microscopic evaluation of the mucosa during endoscopy procedures, CLE is of great significance in the diagnosis and surveillance of early cancer in the upper gastrointestinal tract. Specifically, it has the following advantages: CLE can minimise the number of biopsies while ensuring a high diagnostic sensitivity rate, thereby reducing the risk of mucosal damage, bleeding, infection, and other complications caused by repeated biopsies. Therefore, CLE is more appropriate for the long-term monitoring and follow-up of early cancer. In addition, CLE can be used as an important complement to other imaging techniques and compensate for its shortcomings, such as the contrast limitation of HDWLE, low resolution of NBI, and different diagnostic criteria of FICE, thus showing its characteristic superiority in combination diagnosis with other imaging techniques. In addition, CLE can help clinicians make rapid clinical decisions during endoscopy and reduce the delay of clinical intervention due to the time-consuming nature of pathology tests. Finally, CLE has advantages in the boundary determination of early gastrointestinal neoplasms, especially for flat early gastric cancer, which will facilitate effective endoscopic treatment of early tumours.

However, CLE still has many limitations that limit its routine clinical practice. First, CLE cannot observe the entire esophageal or stomach lumen because of the limited field of vision. Moreover, CLE is not able to show the nuclear structure, which makes it difficult to correctly distinguish the differentiation of early cancer. Therefore, it is impossible to use this method alone for early cancer screenings. Second, the high cost and special training required for image interpretation have hindered its widespread use. Therefore, the application of CLE can only be confined to a small number of tertiary hospitals and cannot be extended to the community. Finally, most countries have not reached an agreement on some issues, such as the standard of care indications and physician image interpretation training, the role of the pathologist, codification, and reimbursement [30,71].

## 5. Perspectives 

Despite some limitations, CLE is a novel and promising imaging method for detecting upper digestive tract tumours. More technological innovations and combined applications with other novel detection techniques will help compensate for its inherent defects and further promote the precise diagnosis of early gastrointestinal cancer. With the further discovery of single-cell omics, more molecular characteristics have been explored at different stages of the progression of digestive tract cancers [72]. These molecular markers will be helpful in the diagnosis and differentiation of early-stage cancers. In addition, some studies have shown that the development of digestive tract tumours is accompanied by changes in the body’s microbiota, and early detection of these changes may enable achieve early identification of digestive tract tumours [73]. The efficient combination of the microscopic detection features of CLE and the detection of these biomarkers may provide a new vision for the diagnosis of early cancer in the future. The rapid development of artificial intelligence may bring new hope for innovation in endoscopy technology. Computer-assisted diagnosis (CAD) based on deep learning may be applied in CLE detection of early cancers to further improve its diagnostic accuracy and observer consistency [74]. These are innovative points of future work. Finally, there is an urgent need for internationally standardised image interpretation protocol and application guidelines for CLE [60]. Future studies should also focus on the cost-effectiveness analysis, routine clinical availability, and learning curve optimisation of CLE.

## 6. Conclusions

CLE alone or in combination with other imaging methods is beneficial for enhancing the detection rate and avoiding excessive biopsy in the management of Barrett’s esophagus and ESCN, which enables doctors to conduct clinical intervention as early as possible to improve the prognosis. CLE is superior for diagnosing precancerous lesions and early gastric cancer in some special cases. Even with some shortcomings, more technical innovations and standardisations of CLE will make it more responsive to the needs of routine screening for early upper gastrointestinal cancer.

## Figures and Tables

**Figure 1 cancers-15-00776-f001:**
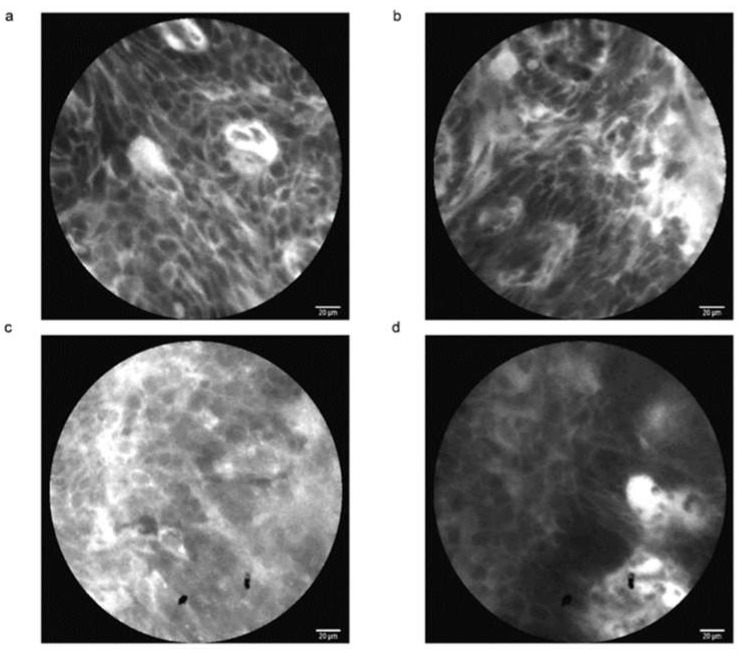
Examples of CLE images in the human esophagus. CLE showed high-grade dysplasia with irregularly thickened epithelial borders and dilated irregular vessels (**a**,**b**). CLE showed inhomogeneous squamous epithelium with irregular architecture and unclear cell boundaries in the early esophageal squamous cell neoplasm (**c**,**d**).

**Figure 2 cancers-15-00776-f002:**
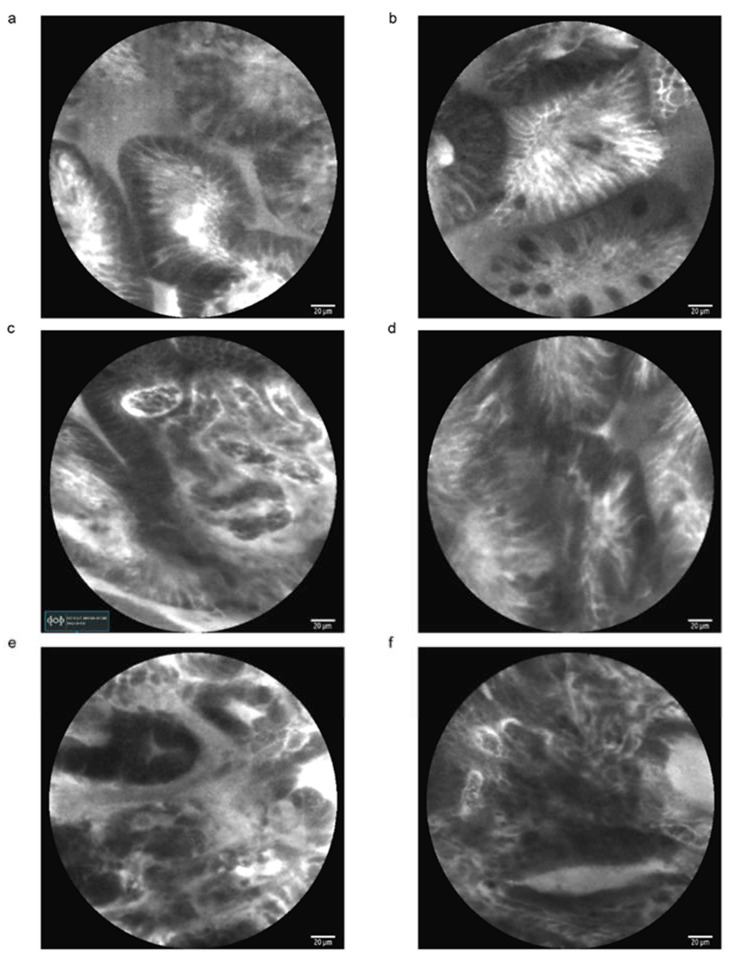
Examples of CLE images in the human stomach. CLE showed glandular atrophy with decreased gastric pits and markedly dilated opening (**a**). CLE showed gastric intestinal metaplasia with dark goblet cells within the columnar epithelium (**b**). CLE showed high-grade intraepithelial neoplasia with distorted pits and irregular epithelial lining in cardia of the stomach (**c**,**d**). CLE showed moderately differentiated adenocarcinoma with atypical/naive glands and dark epithelium (**e**,**f**).

**Table 1 cancers-15-00776-t001:** Meta-analyses of Barrett’s esophagus.

Author and Year of Publication	Classification	Number	Sens (%)	Spec (%)	Positive Likelihood Ratios	Negative Likelihood Ratios
Gupta A (2014) [29]	Per-patient	4	86	83	5.61	0.21
	Per-lesion	7	68	88	6.56	0.24
Wu J (2014) [27]	Per-patient	4	89	75	3.54	0.16
	Per-lesion	7	70	91	9.16	0.27
Xiong YQ (2016) [28]	Per-patient	7	89	83	6.54	0.17
	Per-lesion	10	77	89	8.62	0.23
Xiong YQ (2018) [26]	Per-patient	/	/	/	/	/
	Per-lesion	4	72.3	83.8	4.70	0.30
Thosani N (2016) [25]	pCLE	3	90.3	77.3	/	/
	eCLE	2	90.4	92.7	/	/

Abbreviations: Sens, sensitivity; Spec, specificity.

**Table 2 cancers-15-00776-t002:** Diagnostic value of CLE in the esophagus.

First Author/Year		Study Design	No. of Patients (Lesions)	Intervention Method	Diagnostic Criteria of CLE	Pathological Classification	Per-Patients		Per-Biopsy/Per-Lesion	
							Sens (%)	Spec (%)	Sens (%)	Spec (%)
Canto/2014 [16]		P	192 (1371)	HDWLE + eCLE	Mainz classification	BE-related neoplasia	95	92	86	93
Bertani/2013 [19]		P	100 (/)	HDWLE + pCLE	Miami classification	BE-related dysplasia	100	83	/	/
Shah/2018 [20]		P	66 (/)	Real-time pCLE	Miami classification	HGD/cancer	67	98	/	/
						LGD	60	87	/	/
Sharma/2011 [18]		P	101 (874)	HDWLE or pCLE	/	HGD/EAC	93.5	67.1	68.3	87.8
Pietro/2015 [31]		P	55 (194)	AFI + pCLE	New BE criteria	HGD/IMC	100	53.6	100	67.1
						LGD + HGD/IMC	96.4	74.1	83.3	72.5
Richardson/2019 [21]		P	172 (/)	pCLE	Miami classification	IM	/	/	/	/
Krajciova/2020 [22]		P	56 (/)	pCLE	Miami classification	IM after endoscopic treatment	100	100	/	/
Ghatwary/2019 [23]		P	96 (/)	pCLE	Automatic stage classification	IM	97	96	/	/
						neoplasia mucosa	94	97	/	/
Pietro/2019 [24]		P	57 (/)	pCLE	New criteria in phase I	LGD	/	/	81.9	74.6
Huang/2015 [37]		P	52 (56)	chromoendoscopy-guided CLE	/	ESCN	/	/	95.7	90.0
Piyapan/2018 [39]		P	24 (34)	LCE + pCLE	Cellular and vascular criteria	ESCN	/	/	80	67
Prueksapanich/2015 [38]		P	44 (21)	LCE + pCLE	Cellular and vascular criteria	ESCN	/	/	83	92
Li/2015 [40]		P	64 (64)	pCLE	SMS criteria	Esophageal squamous cell intraepithelial neoplasia	/	/	81	90.7
Guo/2015 [41]		P	356 (117)	I-Scan + pCLE	SMS criteria	ESCN	/	/	74.5	92.9

Abbreviations: P, prospective; Sens, sensitivity; Spec, specificity.

**Table 3 cancers-15-00776-t003:** Diagnostic value of CLE in stomach.

First Author/Year	Study Design	No. of Patients (Lesions)	Intervention Method	Diagnostic Criteria of CLE	Pathological Classification	Real-Time		Offline	
						Sens (%)	Spec (%)	Sens (%)	Spec (%)
Zhang/2008 [44]	P	132	eCLE	Gastric pit patterns by Zhang et al.	AG			83.6	99.6
Li/2016 [45]	P	244	pCLE	New pCLE classification by Li et al.	AG	88.51	99.19	89.86	99.25
					IM	92.34	99.34	93.69	99.40
					LGIN			84.0	99.78
					HGIN			88.89	99.89
Liu/2015 [46]	P	87 (253)	CLE	Gastric pit patterns by Zhang et al.	AG	92.31	86.18	/	/
					Metaplastic atrophic gastritis	91.94	96.86		
Yu/2019 [48]	P	431 (431)	pCLE	New diagnostic criteria in phase I	AG	90.3	78.8		
Lim/2013 [49]	P	20 (125)	pCLE	GIM diagnostic criteria by Lim et al.	GIM	90.9	84.7	95.5	94.9
Li/2014 [51]	P	168 (131)	eCLE + targeted biopsies	GIM diagnostic criteria by Guo et al.	GIM and GIN	/	/	91.67	96.77
Chen/2018 [52]	R	322	pCLE	2011 Miami classification and Qilu classification for gastric superficial lesions	AG and/or IM	/	/	86.8	81.8
					Intraepithelial neoplasia	/	/	96.3	87.1
Ma/2019 [53]	R	119 (154)	pCLE	AG diagnostic criteria by Liu	AG	/	/	94.34	91.09
				GIM diagnostic criteria by Guo et al. and Dixon et al.	GIM	/	/	84.47	92.16
				LGIN diagnostic criteria by Li et al.	LGIN	/	/	85.29	87.50
				HGIN diagnostic criteria by Li et al.	HGIN	/	/	95.83	97.17
Chu/2021 [54]	R	226	pCLE	2011 Miami classification	GIM	/	/	85.55	80.28
					GIN	/	/	65.11	92.78
					EGC	/	/	100	94.55
					GIN + EGC	/	/	86.05	92.77
Sun/2020 [55]	P	47	pCLE	Cresyl violet staining characteristics	GIM	91.95	93.51	/	/
Pittayanon/2013 [56]	P	45 (59)	FICE + pCLE	GIM diagnostic criteria by Guo et al.	GIM	/	/	96.5	90.5
Zuo/2017 [57]	P	238 (212)	FICE + pCLE	New classification of gastric pit patterns	GIM	95.0	94.6	/	/
					LGIN	87.5	98.0		
Bok/2013 [62]	P	46 (54)	pCLE	Miami classification	Gastric adenocarcinoma	90.6	90.9	87.5	95.5
Li/2011 [63]	P	1572 (/)	eCLE	Two-tiered CLE classification	Gastric superficial cancer/HGIN	88.9	99.3	/	/
Gong/2015 [64]	P	82 (86)	ME-NBI + CLE	Gastric pit patterns classification and 2-tiered classification	Gastric neoplastic lesions	94.44	90.91	/	/
Horiguchi/2018 [65]	P	30 (36)	pCLE	Miami classification	EGC after Hp eradication	/	/	/	/
Park/2017 [66]	P	101 (104)	pCLE	/	EGC	/	/	/	/
Zhang/2018 [69]	R	33	nCLE	nCLE criteria	GIST	/	/	78.6	94.7
					Gastric carcinoma	/	/	100	96.4
Zhang/2019 [70]	P	60	nCLE	nCLE criteria	GIST	73.7	95.1	84.2	95.1
					Gastric carcinoma	83.3	95.8	100	97.9

Abbreviations: P, prospective; R, retrospective; Sens, sensitivity; Spec, specificity.

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
