# Peer review of "Confocal Laser Endomicroscopy for Detection of Early Upper Gastrointestinal Cancer"

_cancers, 2023, doi:10.3390/cancers15030776_

Round 1

Reviewer 1 Report

Dear Editor,

I was happy to evaluate the study " Confocal Laser Endomicroscopy for Detection of Early Upper Gastrointestinal Cancer" by Han et al. Early diagnosis for upper gastrointestinal cancer is very important. A nice compilation presented is helpful to the reader in this regard. With the development of the confocal laser endomicroscopy technique, gastrointestinal cancers can be diagnosed earlier. Thus, survival may be improved and treatment may be easier and cheaper.

Kind regards

Reviewer 2 Report

This manuscript reviewed various clinical studies on Confocal Laser Endomicroscopy (CLE) in early esophageal and gastric cancer. When combined with other imaging methods, CLE can improve the detection of Barrett's esophagus and esophageal squamous cell carcinoma more quickly and prevent excessive biopsy. CLE provides more evidence to support clinical decisions for endoscopists, allowing doctors to intervene early and improve outcomes. Therefore, this work could be accepted after the author complements the content, articulates the obscure content, and corrects the format.

1.      Authors should make in-depth analysis of the advantages and disadvantages of Confocal laser endomicroscopy in early cancer detection.

2.      There are two tables called Table 2, which are not clearly quoted in the manuscript. Please clarify further.

3.      The part of Clinical Problems and Perspectives needs more in it. The authors are suggested to highlight important problems and include in-depth thinking about the future development of this work.

4.      For future works, it is suggested that authors take into consideration the combination of CLE and the recent omics technologies used in early gastric cancer detection, e.g. single-cell omics [PMID: 31067475], and microbiome [PMID: 30478535]…

5.      The format needs to be unified. For example, the formats of “p=” and “p = ” and the spaces before and after "+" in Table 2 should be unified.

6.      The format errors need to be corrected. For example, missing “.” after "et al" and "vs".

Round 2

Reviewer 2 Report

The revision is acceptable.